# Unprecedented Elimination Reactions of Cyclic Aldols: A New Biosynthetic Pathway toward the Taiwaniaquinoid Skeleton

**DOI:** 10.3390/molecules28041524

**Published:** 2023-02-04

**Authors:** Juan J. Guardia, Antonio Fernández, José Justicia, Houda Zentar, Ramón Alvarez-Manzaneda, Enrique Alvarez-Manzaneda, Rachid Chahboun

**Affiliations:** 1Departamento de Química Orgánica, Facultad de Ciencias, Instituto de Biotecnología, Universidad de Granada, 18071 Granada, Spain; 2Área de Química Orgánica, Departamento de Química y Física, Universidad de Almería, 04120 Almería, Spain

**Keywords:** bioactive natural products, synthesis, diterpenoids, taiwaniaquinoids

## Abstract

The acid treatment of 6,7-seco-abietane dialdehydes gives, in high yield, the corresponding derivatives with the 4a-methyltetrahydrofluorene skeleton of taiwaniaquinoids. A mechanism involving the elimination of formic acid from the cyclic aldol intermediate is proposed here. This process can be postulated as a new biogenetic pathway from abietane diterpenes to taiwaniaquinoids. Using this novel reaction, the first enantiospecific synthesis of bioactive natural cupresol and taxodal has been obtained.

## 1. Introduction

Taiwaniaquinoids belong to a group of terpenoids with an unusually rearranged 5(6→7) or 6-nor-5(6→7)abeo-abietane skeleton, which have been isolated from species of East Asian conifers during the last 20 years [1]. Although little is known about their biological activities, preliminary studies have revealed some interesting properties, including cytotoxic and trypanocidal effects [2,3,4,5,6,7]. The other structurally related compounds are the bioactive seco-taiwaniquinoids cupresol [8] and taxodal [9]. Figure 1 shows some representative examples of this type of compound.

The promising biological activities and the uncommon structural features of such compounds have motivated the development of several synthetic approaches over the past few years. Thus, diverse total syntheses have been reported, including Pd-catalyzed intramolecular reductive cyclization [10,11], a domino intramolecular acylation carbonyl-α-*tert*-alkylation reaction [12], intramolecular Heck cyclization [13], Nazarov cyclization [14], tandem acylation–Nazarov cyclization [15], acid-promoted Friedel–Crafts acylation/alkylation [16,17], cyclization of aryldienes [18,19,20], and thermal ring expansion/4-electrocyclization [21]. Asymmetric syntheses have also been reported for these compounds, including enantioselective Tsuji allylation [22], enantiospecific thermal 6-electrocyclization [23,24], enantioselective Heck reaction [25,26], iridium-catalyzed borylation, and palladium-catalyzed asymmetric α-arylation [27], as well as the enantioselective conjugate addition of arylboronic acids that are catalyzed by palladium [28]. The synthesis of taiwaniaquinoids, starting from abietane diterpenes, deserves a special mention. On the one hand, the abundance of abietane derivatives in their natural sources, such as (-)-abietic acid, means that these raw materials are inexpensive, which lends commercial interest to this approach, due to the few step sequences that are usually involved. Alternatively, the search for a practical means of transforming abietane terpenoids into taiwaniaquinoids could be of interest in regard to establishing possible biogenetic pathways, because of the structural relationship between the two types of metabolites and their simultaneous presence in their natural sources. A few syntheses of taiwaniaquinoids from abietane derivatives, such as 6,7-dehydroferruginol (**1**, R: H) and (-)-abietic acid (**2**) (see Figure 2), which involve the B ring contraction as the key step, have been reported, and biosynthetic pathways have been proposed on the basis of these transformations. Three biosynthetic proposals have been made, which postulate a 6,7-dehydroferruginol (**1**) derivative as the precursor. The pinacol rearrangement of a 6,7-diol derived from **1** could afford the corresponding cyclopentane carboxaldehyde, which is a possible precursor of the C20 taiwaniaquinoids, such as taiwaniaquinone A and D; however, this conjecture, which was postulated by Cheng [29], could not be experimentally confirmed in a recent study [30,31]. Node et al. postulated the transformation of the seco-abietane dialdehyde, resulting from the oxidation of the C6–C7 double bond of compound **1**, into standishinal (see Figure 1), through a Friedel–Crafts-type cyclization, which was experimentally corroborated by these authors [32]. A third proposal involving the benzylic acid rearrangement of hydroxydione **3** induced by an intramolecular nucleophilic attack has also been supported experimentally by Gademann et al. [33,34]. More recently, our group reported the synthesis of C20 taiwaniaquinoids, such as taiwaniaquinone A, and a wide variety of existing taiwaniaquinoids, based on the cleavage of the C6–C7 double bond of the abietane diterpenes that are derived from (-)-abietic acid (**2**), via the hydroxyaldehyde intermediate **4**, and proposed this process as the key step in a possible biosynthetic pathway [30,31]. Recently, two syntheses, which use a Wolff-type reaction in order to achieve the B ring contraction of the abietane skeleton, have been reported [34,35].

## 2. Results and Discussion

Continuing our research on the synthesis of taiwaniaquinoids from abietane diter-penes, and the related biosynthetic pathway, and after our unsuccessful attempt to achieve the B ring contraction via pinacol rearrangement [30], we focused on the trans-formation of 6,7-dehydroabietane derivatives, which are similar to compound **1**, into taiwaniaquinoids. Thus, we considered the oxidative degradation of the C6–C7 double bond and the subsequent acid-mediated cyclization of the resulting seco-abietane dialdehyde, through a Friedel–Crafts-type reaction process, as reported by Node [32], or via an intramolecular aldol condensation, providing the desired B ring contraction (Figure 1).

In a previous study by Node et al. [32], the seco-abietane dialdehyde resulting from the reductive ozonolysis of the 6,7-dehydroderivative was isolated and, after the deprotection of the phenolic hydroxyl group, was subjected to acid treatment, affording standishinal as the sole product; this compound results from a Friedel–Crafts-type cyclization, which is probably due to the activation of the aromatic ring by the hydroxyl group. Several Brønsted and Lewis acids were assayed, and (+)-camphorsulfonic acid was found to provide the best results [32].

Taking into account these antecedents, seco-abietane dialdehyde **5** was synthesized and its behavior under the previously reported acidic conditions was investigated. First, the reaction with camphorsulfonic acid (1.1 equiv.) was assayed at room temperature for 14 h, following the procedure of Node et al. [32]. Surprisingly, standishinal (**8**) (5%), which was described by Node et al. as the sole product of this reaction, was obtained as the minor product, together with phenol **6** (80%), and aldehyde **7** (7%). Compound **7**, which is the dehydration product of standishinal (**8**), has the characteristic 4a-methyltetrahydrofluorene skeleton of C_20_ taiwaniaquinoids, such as taiwaniaquinone H. This interesting result encouraged us to investigate the behavior of dialdehyde **5** under different acid conditions (Table 1). Some conclusions can be obtained from these results. Thus, the use of trifluoroacetic acid (entry two) favored the formation of C_20_ compound **7**, resulting from a Friedel–Crafts cyclization. The C_19_ compound **6** is the major product when camphorsulfonic acid, which is a less strong acid, is used (entry one). Under weaker acid conditions (entries three and four), compound 6 was the only product. Treatment with BF_3_·Et_2_O afforded a 1:1 mixture of C_19_ and C_20_ compounds **6** and **7** (entry five). Moreover, a mixture of these compounds, in which C_20_ terpenes were predominant, was obtained when metal triflates were used (entries six to eight).

To establish the scope and the limitations of this procedure, the behavior of a series of seco-abietane dialdehydes [36] with different functionalities in the gem-dimethyl group and in the aromatic ring was investigated (Table 2).

Some conclusions can be drawn from the results that are depicted in Table 2. Phenol dialdehydes, such as compound **5** (Table 1), produced significant amounts of C_20_ compounds, such as compounds **7** and **8**, due to the activation of the aromatic ring in the Friedel–Crafts cyclization (Figure 2). The less activated substrates, such as dialdehyde **9** (entry one) or aromatic ethers, only gave the corresponding C_19_ compound (entries one to six). Dialdehyde **18**, with an ester group in the cyclohexane ring, yielded, in the presence of camphorsulfonic acid, the unsaturated lactone **19**, resulting from the intramolecular attack of the enol aldehyde on the ester group (entry seven). When milder acidic conditions were used, the corresponding C_19_ compound **20**, with lactone **19**, was obtained (entry eight). On the other hand, dialdehyde **21** showed a different behavior, due to the tendency of the acetyloxymethyl group to undergo the intramolecular attack of the enol aldehyde, leading to a dihydrofuran derivative (entries 10 and 11). When camphorsulfonic acid was used, the starting material **21** remained unaltered, which was probably due to the reversion of aldolic condensation (entry nine).

A possible mechanism is shown in Figure 3. The cyclopentane ring is formed after the attack of the aliphatic aldehyde enol on the aromatic aldehyde. The dehydration of the protonated resulting aldol (**I**) leads to benzylic cation **II**, which is finally converted into the tricyclic cyclopentene derivative **6** after losing formic acid.

The results of the further experiments were consistent with the above proposal. Thus, the treatment of aldol **24**, which was easily obtained from dialdehyde **11**, with amberlyst A-15, gave compound **12** in high yield (Figure 4).

Another experiment that confirms the proposed mechanism that is depicted in Figure 5. The O-methyl aldol **26** was obtained when dialdehyde **11** was treated with sodium methoxide in methanol, until the disappearance of the starting material, and then with 2N hydrochloric acid. Compound **26** was formed after the nucleophilic attack of methanol on intermediate **III**, which is similar to intermediate **II**, as postulated in Figure 3. The further treatment of compound **26** with cationic resin produced compound **12**.

In accordance with the proposed mechanism, the acid was not consumed during the reaction. This outcome led us to believe that cyclization could occur using a substoichiometric quantity of acid. Compound **12** was effectively obtained in a 90% yield after the treatment of dialdehyde **11** with 30 mol % of camphorsulfonic acid (Figure 6).

In view of the above results, the transformation of the abietane skeleton into the 4a-methyltetrahydrofluorene skeleton, which is typical of some taiwaniaquinoids, in only two steps, is feasible. Therefore, this process could be proposed as a new biogenetic pathway from abietane diterpenes to taiwaniaquinoids. The scope and the limitations of this unprecedented cyclization are currently under study in order to apply it in organic synthesis.

This rearrangement of seco-abietane dialdehydes has been employed in order to achieve the first enantiospecific synthesis of bioactive taxodal (**28**) and cupresol (**30**) from methyl ether **12**, which is easily obtained from (-)-abietic acid (Figure 7) [37].

The treatment of ketoaldehyde **27**, resulting from the ozonolysis of ether **12**, with ti-ophenol in basic media, gave bioactive taxodal (**28**) [37]. Alternatively, the oxidation of compound **27** with sodium chlorite produced a 1:1 mixture of epimers **29**. The deprotection of methyl ether and further chromatography allowed us to obtain the enantiopure bioactive cupresol (**30**) [37,38].

## 3. Materials and Methods

### 3.1. Materials

Unless otherwise stated, the reactions were performed in oven-dried glassware under an argon atmosphere using dry solvents. The solvents were dried as follows: THF over Na–benzophenone and CH_2_Cl_2_ and MeOH over CaH_2_.

Thin-layer chromatography (TLC) was performed using Merck silica gel 60-F254 precoated plates (0.25 mm) and visualized by UV fluorescence quenching and phosphomolybdic acid solution staining. Flash chromatography was performed on silica gel (Merck Kieselgel 60, 230–400 mesh). Chromatography separations were carried out with a conventional column on silica gel 60 (230–400 Mesh), using Hexane–t-BuOMe (H–E) mixtures of increasing polarity.

^1^H and ^13^C NMR spectra were recorded on a Varian instrument (at 500 MHz and 125 MHz, respectively). CDCl_3_ was treated with K_2_CO_3._ The data for ^1^H NMR spectra are reported as follows: chemical shift (δ ppm) (multiplicity, coupling constant (Hz), and integration), with the abbreviations s, br s, d, br d, dd, ddd, dd br t, q, and m denoting singlet, broad singlet, doublet, broad doublet, double doublet, double double doublet, broaddouble doublet, triplet, quartet, and multiplet, respectively. *J* = coupling constant in Hertz (Hz).

Infrared spectra (IR) were recorded as thin films or as solids on a Perkin Elmer model One FTIR spectrophotometer with samples between sodium chloride plates or as potassium bromide pellets and are reported in frequency of absorption (cm^−1^). The ([α]_D_) measurements were carried out in a Perkin Elmer 341 polarimeter, using a 1 dm length cell and CHCl_3_ as a solvent. The concentration is expressed in mg/mL. HRMS were recorded on an AutoSpecQ VG-Analytical (Fisons) spectrometer, using FAB with thioglicerol or glycerol matrix doped in NaI 1%.

### 3.2. Methods

#### 3.2.1. General Procedure for the Obtention of Seco-Abietane Dialdehydes

Seco-abietane dialdehydes were obtained after the reductive ozonolysis of the corresponding 6,7-dehydro-8,11,13-abietatriene diterpenes, using the following procedure:

An O_3_/O_2_ mixture was slowly bubbled through a solution of the selected 6,7-dehydro-8,11,13-abietatriene derivative (1.0 mmol) in dry CH_2_Cl_2_ (15 mL) at −78 °C, and the course of the reaction was monitored by TLC. When the TLC showed no starting material, the solution was flushed with argon, triphenylphosphine (1.1 mmol) was added, and the mixture was stirred at room temperature for 4 h. Then, the solvent was removed, affording a crude product, which was directly purified with a flash chromatography column on silica gel (*t*-BuOMe -Hexane mixtures) to produce the expected dialdehyde.

#### 3.2.2. General Procedure for the Reaction of Seco-Abietane Dialdehydes with (+)-CSA, Amberlyst A-15, Bi(OTf)_3_, Sc(OTf)_3_, or Gd(OTf)_3_

The selected acid (0.12 mmol) was added to a solution of the corresponding dialdehyde (0.1 mmol) in dry CH_2_Cl_2_ (10 mL), the mixture was stirred at room temperature for the specified time (see Table 1), and the course of the reaction was monitored by TLC. When the starting material was consumed, the solvent was removed under vacuum conditions, and the crude product was directly purified with a flash chromatography column on silica gel (Hexane–*t*-BuOMe mixtures) to produce the corresponding final product(s).

#### 3.2.3. General Procedure for the Reaction of Seco-Abietane Dialdehydes with H_2_SO_4_, CF_3_CO_2_H, HCO_2_H, or BF_3_·Et_2_O

The indicated acid (0.12 mmol) was added to a solution of the corresponding dialdehyde (0.1 mmol) in dry CH_2_Cl_2_ (10 mL) and the mixture was stirred at the specified temperature for the indicated time (see Table 2). The course of the reaction was monitored by TLC. When the starting material was consumed, the reaction mixture was diluted with CH_2_Cl_2_ (30 mL) and washed with a sat. aqueous solution of NaHCO_3_ (2 × 10 mL) and brine (2 × 10 mL), and the organic phase was dried over anhydrous Na_2_SO_4_. Removal of the solvent under vacuum conditions afforded a crude product, which was purified with a flash chromatography column on silica gel (Hexane– *t*-BuOMe mixtures) to produce the corresponding final product(s).

#### 3.2.4. Experimental Procedures

(R)-7-Isopropyl-1,1,4a-trimethyl-2,3,4,4a-tetrahydro-1*H*-fluoren-6-ol (**6**). Yellow syrup (30% *t*-BuOMe/hexanes). [α]_D_^25^= + 28.7 (c = 0.15 CHCl3). 1H NMR (CD_3_COCD_3_, 500 MHz) δ: 0.91 (ddd, *J* = 13.0, 13.0, 3.7 Hz, 1H), 1.06 (ddd, *J* = 13.2, 13.2, 4.0 Hz, 1H), 1.20 (d, *J* = 6.9 Hz, 3H,), 1.21 (s, 3H), 1.21 (d, *J* = 6.9 Hz, 3H), 1.25 (s, 3H), 1.29 (s, 3H), 1.58–1.65 (m, 2 H), 1.93–1.98 (m, 1H), 2.01–2.10 (m, 1H), 3.31 (h, *J* = 6.9 Hz, 1H), 6.26 (s, 1H), 6.76 (s, 1H), 7.06 (s, 1H). ^13^C NMR (CD_3_COCD_3_, 125 MHz) δ: 21.1 (CH_2_), 23.7 (CH_3_), 23.8 (CH_3_), 24.7 (CH_3_), 26.5 (CH), 28.2 (CH_3_), 32.3 (CH_3_), 36.5 (CH), 39.6 (CH_2_), 44.1 (CH_2_), 51.9 (C), 110.1 (CH), 119.2 (C), 122.2 (CH), 133.7 (CH), 135.4 (C), 153.2 (C), 155.2 (C), 161.6 (C). IR (film): 3363, 1465, 1428, 1286, 1226, 1171, 1007, 887, 759, 669 cm^−1^. HRMS (FAB) *m*/*z* calcd for C_19_H_26_ONa (M + Na^+^) 293.1881, found 293.1893.

(S)-8-hydroxy-7-isopropyl-1,1,4a-trimethyl-2,3,4,4a-tetrahydro-1*H*-fluorene-5-carbaldehyde (**7**). Yellow syrup (15% *t*-BuOMe/hexanes). [α]_D_^25^= + 14.6 (c = 0.15 CHCl_3_). ^1^H NMR (CDCl_3_, 500 MHz) δ: 1.12–1.24 (m, 2 H), 1.28 (s, 3 H), 1.31 (d, *J* = 6.9 Hz, 3 H), 1.32 (d, *J* = 6.9 Hz, 3 H), 1.34 (s, 3 H), 1.54 (s, 3 H), 1.65–1.71 (m, 2 H), 1.92–2.01 (m, 1 H), 2.72–2.78 (m, 1 H), 3.22 (h, *J* = 6.9 Hz, 1 H), 5.31 (br s, 1 H), 6.45 (s, 1 H), 7.55 (s, 1 H), 10.24 (s, 1 H). ^13^C NMR (CDCl_3_, 125 MHz) δ: 19.5 (CH_2_), 22.3 (CH_3_), 22.6 (CH_3_), 22.7 (CH_3_), 25.5 (CH_3_), 26.9 (CH), 31.6 (CH_3_), 35.7 (C), 36.6 (CH_2_), 42.2 (CH_2_), 51.9 (C), 114.4 (CH), 125.5 (C), 126.8 (CH), 129.6 (C), 132.7 (C), 150.7 (C), 156.6 (C), 165.0 (C), 190.6 (CH). IR (film): 3331, 1660, 1606, 1559, 1468, 1270, 1217, 1172, 1156, 779 cm^−1^. HRMS (FAB) *m*/*z* calcd for C_20_H_26_O_2_Na (M + Na^+^) 321.1830, found 321.1822.

2-((1S,2S)-2-formyl-1,3,3-trimethylcyclohexyl)-5-isopropylbenzaldehyde (**9**). Colorless oil (10% *t*-BuOMe/hexanes). ^1^H NMR (CDCl_3_, 500 MHz) δ: 0.93 (s, 3H), 1.20–1.25 (m, 3H), 1.26 (s, 3H), 1.26 (d, *J* = 6.9 Hz, 3H), 1.27 (d, *J* = 6.9 Hz, 3H), 1.48 (s, 3H), 1.58–1.71 (m, 3H), 2.31–2.36 (m, 1H), 3.27 (h, *J* = 6.9 Hz, 1H), 7.11 (s, 1H), 7.83 (s, 1H), 7.91 (s, 1H), 9.85 (d, *J* = 3.8 Hz, 1H), 10.46 (s, 1H). ^13^C NMR (CDCl_3_, 125 MHz) δ: 13.4 (CH_3_), 18.9 (CH_2_), 19.7 (CH_2_), 24.0 (CH_2_), 22.23 (CH_3_), 22.26 (CH_3_), 29.8 (C), 33.5 (CH), 49.5 (CH), 59.3 (CH), 110.8 (CH), 126.5 (CH), 127.4 (CH), 132.4 (C), 150.9 (C), 158.6 (C), 191.5 (CH), 206.9 (CH). HRMS (FAB) *m*/*z* calcd for C_20_H_28_O_2_Na (M + Na^+^) 323.1987, found 323.1999.

(R)-7-isopropyl-1,1,4a-trimethyl-2,3,4,4a-tetrahydro-1H-fluorene (**10**). Colorless syrup (Hexane). [α]_D_^25^= + 25.4 (c = 0.15, CHCl_3_). ^1^H NMR (CDCl_3_, 500 MHz) δ: 0.95–1.05 (m, 1H), 1.09–1.15 (m, 1H), 1.27 (d, *J* = 6.9 Hz, 3H), 1.28 (d, *J* = 6.9 Hz, 3H), 1.28 (s, 3H), 1.31 (s, 3H), 1.38 (s, 3H), 1.58–1.69 (m, 2H), 1.90–2.00 (m, 1H), 2.15 (dd, *J* = 12.8, 1.6 Hz, 1H), 2.93 (h, *J* = 6.9 Hz, 1H), 6.36 (s, 1H), 7.01 (d, *J* = 7.3 Hz, 1H), 7.17 (d, *J* = 7.3 Hz, 1H), 7.18 (s, 1H). ^13^C NMR (CDCl_3_, 125 MHz) δ: 19.8 (CH_2_), 20.3 (CH_3_), 20.32 (CH_3_), 23.5 (CH_3_), 25.3 (CH_3_), 31.3 (CH_3_), 34.1 (CH), 38.1 (CH_2_), 42.7 (CH_2_), 50.9 (C), 118.4 (CH), 120.1 (CH), 120.8 (CH), 122.2 (CH), 142.2 (CH_3_), 147.0 (CH), 152.2 (C), 164.2 (C). IR (film): 1615, 1478, 1480, 1382, 1260, 1091, 1027, 885, 819,672, 641 cm^−1^. HRMS (FAB) *m*/*z* calcd for C_19_H_26_Na (M + Na^+^) 277.1932, found 277.1920.

(R)-7-isopropyl-6-methoxy-1,1,4a-trimethyl-2,3,4,4a-tetrahydro-1H-fluorene (**12**). Colorless syrup (5% *t*-BuOMe/hexanes). [α]_D_^25^= +7.9 (c = 0.45, CHCl_3_). ^1^H NMR (CDCl_3_, 500 MHz) δ: 1.04 (ddd, *J* = 13.2, 13.2, 3.7 Hz, 1 H), 1.13 (ddd, *J* = 14.0, 14.0, 4.5 Hz, 1 H), 1.24 (d, *J* = 6.9 Hz, 3 H), 1.25 (d, *J* = 6.9 Hz, 3 H), 1.26 (s, 3 H), 1.31 (s, 3 H), 1.39 (s, 3H), 1.61–1.69 (m, 2 H), 1.98 (dddt, *J* = 13.7, 13.7, 13.7, 3.5 Hz, 1 H), 2.13 (dd br, *J* = 12.7, 1.5 Hz, 1 H), 3.35 (h, *J* = 6.9 Hz, 1 H), 3.87 (s, 3 H), 6.33 (s, 1 H), 6.81 (s, 1 H), 7.16 (s, 1 H). ^13^C NMR (CDCl_3_, 125 MHz) δ: 20.0 (CH_2_), 23.2 (CH_3_), 23.2 (CH_3_), 23.9 (CH_3_), 25.6 (CH_3_), 26.9 (CH), 31.4 (CH_3_), 35.6 (C), 38.4 (CH_2_), 42.9 (CH_2_), 51.2 (C), 56.1 (CH_3_), 104.8 (CH), 117.9 (CH), 120.5 (CH), 134.7 (C), 135.1 (C), 153.8 (C), 154.6 (C), 162.0 (C). IR (film): 1483, 1464, 1416, 1309, 1221, 1199, 888, 669 cm^−1^. HRMS (FAB) *m*/*z* calcd for C_20_H_28_ONa (M + Na^+^) 307.2039, found 307.2047.

((1R,2R,3S)-2-formyl-3-(2-formyl-4-isopropyl-3-methoxyphenyl)-1,3-dimethyl cyclohexyl) methyl acetate (**14**). Colorless syrup (15% *t*-BuOMe/hexanes). ^1^H NMR (CDCl_3_, 500 MHz) δ: 1.06 (s, 3H), 1.23 (d, *J* = 6.9 Hz, 3 H), 1.26 (d, *J* = 6.9 Hz, 3 H), 1.48 (s, 3H), 1.55 (dd, *J* = 6.0, 5.7 Hz, 1 H), 1,66–1.75 (m, 2 H), 2.09 (s, 3H), 2.18 (dd, *J* = 13.5, 8.0 Hz, 1H), 3.27 (h, *J* = 6.9 Hz, 1 H), 3.31 (d, *J* = 3.1 Hz, 1H), 3.54 (d, *J* = 11.0 Hz, 1H), 3.71 (d, *J* = 11.0 Hz, 1H), 3.72 (s, 3H), 7.25 (d, *J* = 8.5 Hz, 1H), 7.34 (d, *J* = 8.5 Hz, 1H), 9.75 (d, *J* = 2.9 Hz, 1 H), 10.74 (s, 1H). ^13^C NMR (CDCl_3_, 125 MHz) δ: 18.6 (CH_2_), 20.8 (2 x CH_3_), 23.2 (2 x CH_3_), 23.8 (CH_3_), 25.7 (CH), 29.7 (CH_2_), 33.5 (CH_2_), 36.5 (C), 40.9 (C), 57.6 (CH), 63.4 (CH_3_), 71.3 (CH_2_), 123.3 (CH), 129.4 (CH), 132.3 (C), 140.2 (C), 145.6 (C), 158.6 (C), 170.8 (C), 197.5 (CH), 204.7 (C). HRMS (FAB) *m*/*z* calcd for C_23_H_32_O_5_Na (M + Na^+^) 411.2147, found 411.2135.

((1R,4aR)-7-isopropyl-8-methoxy-1,4a-dimethyl-2,3,4,4a-tetrahydro-1H-fluoren-1-yl)methyl acetate (**15**). Yellow syrup (10% *t*-BuOMe/hexanes). [α]_D_^25^= + 13.3 (c = 8.1, CHCl_3_). ^1^H NMR (CDCl_3_, 500 MHz) 1.00–1.10 (m, 1 H), 1.13–1.20 (m, 1 H), 1.24 (d, *J* = 6.9 Hz, 3 H), 1.25 (d, *J* = 6.9 Hz, 3 H), 1.30 (s, 3 H), 1.39 (s, 3 H), 1.55–1.65 (m, 2 H), 1.67–1.74 (m, 2 H), 2.12 (s, 3 H), 3.35 (h, *J* = 6.9 Hz, 1 H), 3.88 (s, 3 H), 4.21 (d, *J* = 10.7 Hz, 1 H), 4.26 (d, *J* = 10.7 Hz, 1 H), 6.49 (s, 1 H), 6.98 (d, *J* = 7.7 Hz, 1 H), 7.04 (d, *J* = 7.7 Hz, 1 H). ^13^C NMR (CDCl_3_, 125 MHz) δ: 18.9 (CH_2_), 20.6 (CH_3_), 21.0 (CH_3_), 23.81 (CH_3_), 23.84 (CH_3_), 23.88 (CH_3_), 26.6 (CH), 37.6 (CH_2_), 37.6 (CH_2_), 38.8 (C), 51.1 (C), 61.1 (CH_3_), 73.1 (CH_2_), 116.6 (CH), 117.6 (CH). 122.7 (CH), 133.3 (C), 138.9 (CH), 150.7 (C), 154.5 (C), 158.2 (C), 171.4 (C). IR (film): 1741, 1466, 1372, 1240, 1034, 814 cm^−1^. HRMS (FAB) *m*/*z* calcd for C_22_H_30_O_3_Na (M + Na^+^) 365.2093, found 365.2101.

((1R,2R,3S)-3-(5-(benzyloxy)-2-formyl-4-isopropylphenyl)-2-formyl-1,3-dimethyl cyclohexyl)methyl acetate (**16**). Colorless oil (20% *t*-BuOMe/hexanes). ^1^H NMR (CDCl_3_, 500 MHz) δ: 0. 98 (s, 3H), 1.21 (d *J* = 7.0 Hz, 1H), 1.22 (d *J* = 7.0 Hz, 1H), 1.27 (d, *J* = 6.9 Hz, 3H), 1.28 (d, *J* = 6.9 Hz, 3H), 1.47 (s, 3H), 1.50–1.55 (m, 1H), 1.61–1.68 (m, 1H), 1.70–1.78 (m, 1H), 1.94 (s, 3H), 2.25–2.31 (m, 1H), 3.38 (h, *J* = 6.9 Hz, 1H), 3.45–3.49 (m, 2H), 3.51 (d, *J* = 7.0 Hz, 1H), 5.21 (d, *J* = 12.4 Hz, 1H), 5.26 (d, *J* = 12.4 Hz, 1H), 6.93 (s, 1H), 7.34 (dd, *J* = 8.5, 4.2 Hz, 1H), 7.41 (d, *J* = 4.2 Hz, 4H), 7.87 (s, 1H), 9.86 (s, 1H), 10.60 (s, 1H). ^13^C NMR (CDCl_3_, 125 MHz) δ: 18.6 (CH_2_), 20.6 (CH_3_), 22.31 (2 x CH_3_), 22.33 (2 x CH_3_), 26.5 (CH), 32.8 (CH_2_), 36.1 (C), 36.6 (CH_3_), 40.4 (C), 58.9 (C), 69.9 (CH_2_), 70.5 (CH_2_), 110.5 (CH), 126.8 (2 x CH), 127.5 (C), 128.0 (CH), 128.8 (2 x CH), 135.8 (C), 136.5 (C), 144.2 (C), 159.5 (C), 170.6 (C), 190.7 (CH), 204.5 (CH). HRMS (FAB) *m*/*z* calcd for C_29_H_36_O_5_Na (M + Na^+^) 487.2460, found 487.2474.

((1R,4aR)-6-(benzyloxy)-7-isopropyl-1,4a-dimethyl-2,3,4,4a-tetrahydro-1H-fluoren-1-yl) methyl acetate (**17**). Yellow oil (10% *t*-BuOMe/hexanes). [α]_D_^25^= + 56.6 (c = 20.5, CHCl_3_). ^1^H NMR (CDCl_3_, 500 MHz) δ: 1.02–1.10 (m, 1H), 1.15 (td, *J* = 13.4, 12.8, 4.3 Hz, 1 H), 1.21–1.28 (m, 1 H), 1.24 (d, *J* = 6.9 Hz, 3 H), 1.24 (d, *J* = 6.9 Hz, 3 H), 1.29 (s, 3 H), 1.37 (s, 3 H), 1.65–1.73 (m, 2 H), 1.93–2.02 (m, 2 H), 2.13 (s, 3 H), 4.17 (d, *J* = 10.6 Hz, 1 H), 4.22 (d, *J* = 10.6 Hz, 1 H), 5.10 (s, 2 H), 6.32 (s, 1 H), 6.88 (s, 1H), 7.19 (s, 1 H), 7.33 (t, *J* = 7.4 Hz, 1 H), 7.40 (t, *J* = 7.5 Hz, 2 H), 7.47 (d, *J* = 7.5 Hz, 2 H). ^13^C NMR (CDCl_3_, 125 MHz) δ: 18.9 (CH_2_), 21.0 (CH_3_), 23.02 (CH_3_), 23.05 (CH_3_), 23.8 (CH_3_), 23.9 (CH_3_), 26.9 (CH), 37.0 (CH_2_), 37.7 (CH_2_), 38.8 (C), 51.2 (C), 70.1 (CH_2_), 73.2 (CH_2_), 106.1 (CH), 118.2 (CH), 120.9 (CH), 127.2 (2 x CH), 127.6 (CH), 128.4 (2 x CH_3_), 134.6 (C), 135.7 (C), 137.7 (C), 153.2 (C), 153.8 (C), 156.9 (C), 171.4 (C). IR (film): 1739, 1421, 1308, 1289, 1238,1189, 1039 cm^−1^. HRMS (FAB) *m*/*z* calcd for C_28_H_34_O_3_Na (M + Na^+^) 441.2406, found 441.2399.

(1R,2R,3S)-Methyl 2-formyl-3-(2-formyl-4-isopropyl-5-methoxyphenyl)-1,3-dimethylcyclohexanecarboxylate (**18**). Yellow syrup (20% *t*-BuOMe/hexanes). ^1^H NMR (CDCl_3_, 500 MHz) δ: 1.15 (s, 3 H), 1.19 (d, *J* = 6.9 Hz, 3 H), 1.21 (s, 3 H), 1.48 (s, 3 H), 1.63–1.75 (m, 1 H),1.77–1.87 (m, 2 H), 2.34 (d, *J* = 14.8 Hz, 1 H), 3.03 (s, 3 H), 3.26 (h, *J* = 6.9 Hz, 1 H), 3.90 (s, 3 H), 4.09 (s, 1 H), 6.95 (s, 1 H), 7.82 (s, 1 H), 9.96 (s, 1 H), 10.52 (s, 1 H). ^13^C NMR (CDCl_3_, 125 MHz) δ: 19.8 (CH_2_), 22.1 (2 x CH_3_), 22.4 (2 x CH_3_), 26.2 (CH), 31.7 (CH_2_), 35.9 (CH_2_), 40.9 (C), 43.3 (C), 51.2 (CH), 55.5 (CH_3_), 59.5 (CH_3_), 110.4 (CH), 132.1 (C), 133.1 (C), 134.9 (CH), 147.8 (C), 160.3 (C), 175.5 (C), 191.4 (CH), 203.8 (CH). HRMS (FAB) *m*/*z* calcd for C_22_H_30_O_5_Na (M + Na^+^) 397.1991, found 397.2007.

2-((4R,7aR)-4,7a-dimethyl-1-oxo-1,4,5,6,7,7a-hexahydroisobenzofuran-4-yl)-5-isopropyl-4-methoxybenzaldehyde (**19**). Colorless syrup (20% *t*-BuOMe/hexanes). ^1^H NMR (CDCl_3_, 500 MHz) δ: 1.20 (d, *J* = 6.9 Hz, 6 H), 1.31 (s, 3 H), 1.40 (s, 3 H), 1.95–2.03 (m, 2 H),1.66–1.85 (m, 4 H), 3.27 (h, *J* = 6.9 Hz, 1 H), 3.83 (s, 3 H), 6.06 (s, 1 H), 6.52 (s, 1 H), 7.36 (s, 1 H), 9.81 (s, 1 H). ^13^C NMR (CDCl_3_, 125 MHz) δ: 15.6 (CH_3_), 21.4 (CH_3_), 22.6 (CH_3_), 23.3 (CH_3_), 26.9 (CH_2_), 27.1 (CH_3_), 34.4 (CH_2_), 44.8 (CH), 48.1 (C), 66.6 (C), 82.7 (CH), 103.2 (CH), 124.1 (C), 124.7 (CH), 127.8 (C), 137.6 (C), 150.1 (C), 159.1 (C), 179.9 (C), 199.8 (CH). HRMS (FAB) *m*/*z* calcd for C_21_H_26_O_4_Na (M + Na^+^) 365.1729, found 365.1714.

(1R,4aR)-Methyl 7-isopropyl-6-methoxy-1,4a-dimethyl-2,3,4,4a-tetrahydro-1H-fluorene-1-carboxylate (**20**). Colorless syrup (5% *t*-BuOMe/hexanes). [α]_D_^25^= + 91.4 (c = 0.34, CHCl_3_). ^1^H NMR (CDCl_3_, 500 MHz) δ: 1.20 (d, *J* = 7.3 Hz, 3H), 1.21 (d, *J* = 7.3 Hz, 3H), 1.40 (s, 3H), 1.54 (s, 3H), 1.72–1.77 (m, 2H), 1.79 (d, *J* = 10.5 Hz, 1H), 1.93 (dd, *J* = 13.4, 10.6 Hz, 2H), 2.13 (d, *J* = 12.7 Hz, 1 H), 3.31 (h, *J* = 7.3 Hz, 1 H), 3.81 (s, 3 H), 3.84 (s, 3 H), 6.16 (s, 1 H), 6.78 (s, 1 H), 7.12 (s, 1 H). ^13^C NMR (CDCl_3_, 125 MHz) δ: 18.8 (CH_2_), 21.4 (CH_3_), 22.9 (CH_3_), 23.0 (CH_3_), 24.1 (CH_3_), 26.7 (CH), 36.4 (CH_2_), 37.6 (CH_2_), 47.9 (C), 51.8 (CH_3_), 52.3 (C), 55.8 (CH_3_), 104.3 (CH), 118.3 (CH), 122.8 (CH_3_), 133.0 (C), 135.0 (C), 153.2 (C), 153.7 (C), 154.9 (C), 176.7 (C). HRMS (FAB) *m*/*z* calcd for C_21_H_28_O_3_Na (M + Na^+^) 351.1936, found 351.1941.

((1R,2R,3S)-3-(5-acetoxy-2-formyl-4-isopropylphenyl)-2-formyl-1,3-dimethyl cyclohexyl)methyl acetate (**21**). Colorless syrup (15% *t*-BuOMe/hexanes). ^1^H NMR (CDCl_3_, 500 MHz) δ: 0.98 (s, 3 H), 1.21 (d, *J* = 6.9 Hz, 3 H), 1.22 (d, *J* = 6.9 Hz, 3 H), 1.51 (s, 3 H), 1.59–1.70 (m, 2 H), 1.55–1.59 (m, 5 H), 1.92 (s, 3 H), 2.34 (s, 3 H), 3.02 (h, *J* = 6.9 Hz, 1 H), 3.34 (d, *J* = 4.2 Hz, 1 H), 3.43 (d, *J* = 4.2 Hz, 1 H), 7.15 (s, 1 H), 7.92 (s, 1 H), 9.90 (br s, 1 H), 10.75 (s, 1 H). ^13C NMR (CDCl^_3_, 125 MHz) δ: 18.7 (CH_2_), 20.6 (2 x CH_3_), 20.9 (2 x CH_3_), 22.6 (2 x CH_3_), 27.1 (CH), 32.8 (CH_2_), 36.1 (C), 36.6 (CH_2_), 40.0 (C), 59.5 (CH), 70.2 (CH_2_), 121.4 (CH), 131.8 (CH), 132.6 (C), 139.1 (C), 147.9 (C), 151.9 (C), 168.8 (C), 170.7 (C), 190.7 (CH), 204.0 (CH). HRMS (FAB) *m*/*z* calcd for C_24_H_32_O_6_Na (M + Na^+^) 439.2097, found 439.2105.

2-((4R,7aR)-4,7a-Dimethyl-1,4,5,6,7,7a-hexahydroisobenzofuran-4-yl)-4-hydroxy-5-isopropylbenzaldehyde (**22**). Yellow syrup (40% *t*-BuOMe/hexanes). ^1^H NMR (CDCl_3_, 500 MHz) δ: 1.04 (s, 3 H), 1.10 (s, 3 H), 1.25 (d, *J* = 6.9 Hz, 3 H), 1.27 (d, *J* = 6.9 Hz, 3 H), 1.42–1.50 (m, 2 H), 1.55–1.59 (m, 2 H), 1.64–1.68 (m, 1 H), 1.69–1.78 (m, 1 H), 3.15 (h, *J* = 6.9 Hz, 1 H), 3.20 (d, *J* = 8.5 Hz, 1 H), 3.24 (d, *J* = 8.5 Hz, 1 H), 5.02 (br s, 1 H), 6.07 (s, 1 H), 6.42 (s, 1 H), 7.22 (s, 1 H), 9.80 (s, 1 H). ^13C NMR (CDCl^_3_, 125 MHz) δ: 18.3 (CH_2_), 22.5 (CH_3_), 22.6 (CH_3_), 25.7 (CH_3_), 27.3 (CH), 31.2 (CH_3_), 32.7 (CH_2_), 35.1 (CH_2_), 43.7 (C), 47.1 (C), 69.6 (C), 76.0 (CH_2_), 84.4 (CH), 106.7 (CH), 123.1 (CH), 131.9 (C), 134.8 (C), 148.4 (C), 154.1 (C), 200.2 (CH). IR (film): 1672, 1601, 1381, 1269, 1242, 932 cm^−1^. HRMS (FAB) *m*/*z* calcd for C_20_H_26_O_3_Na (M + Na^+^) 337.1780, found 337.1765.

5-((4R,7aR)-4,7a-Dimethyl-1,4,5,6,7,7a-hexahydroisobenzofuran-4-yl)-4-formyl-2-isopropylphenyl acetate (**23**). Yellow syrup (20% *t*-BuOMe/hexanes). [α]_D_^25^= –7.7 (c = 0.10, CHCl_3_). ^1^H NMR (CDCl_3_, 500 MHz) δ: 1.07 (s, 3 H), 1.10 (s, 3 H), 1.22 (d, *J* = 6.9 Hz, 3 H), 1.23 (d, *J* = 6.9 Hz, 3 H), 1.46 (ddd, *J* = 13.0, 13.0, 4.0 Hz, 1 H), 1.56–1.60 (m, 1 H), 1.63–1.69 (m, 1 H), 1.74 (ddd, *J* = 13.6, 13.6, 3.7 Hz, 1 H), 2.21 (ddd, *J* = 8.8, 8.8, 4.7 Hz, 1 H), 2.33 (s, 3 H), 2.99 (h, *J* = 6.9 Hz, 1 H), 3.21 (d, *J* = 8.4 Hz, 1 H), 3.24 (d, *J* = 8.4 Hz, 1 H), 6.08 (s, 1 H), 6.68 (s, 1 H), 7.35 (s, 1 H), 9.80 (s, 1 H). ^13^C NMR (CDCl_3_, 125 MHz) δ: 18.2 (CH_2_), 21.0 (CH_3_), 22.9 (2 x CH_3_), 25.6 (CH_3_), 27.7 (CH), 31.1 (CH_3_), 32.6 (CH_2_), 35.0 (CH_2_), 43.8 (C), 47.1 (C), 69.6 (C), 76.4 (CH_2_), 84.2 (CH), 111.4 (CH), 123.5 (CH), 137.4 (C), 140.3 (C), 148.2 (C), 149.3 (C), 169.5 (C), 199.8 (CH). IR (film): 1760, 1721, 1462, 1369, 1164, 1025, 757 cm^−1^. HRMS (FAB) *m*/*z* calcd for C_22_H_28_O_4_Na (M + Na^+^) 379.1885, found 379.1896.

(4aR,9S,9aS)-9-hydroxy-7-isopropyl-6-methoxy-1,1,4a-trimethyl-2,3,4,4a,9,9a-hexahydro-1H-fluorene-9a-carbaldehyde (**24**). *t*-BuOK (44 mg, 0.39 mmol) was added to a solution of **11** (107 mg, 0.32 mmol) in THF (3 mL) at 0 °C, and the reaction mixture was stirred at 0 °C for 15 min, at which time TLC showed no starting material. Then, water (1 mL) was added to quench the reaction and the mixture was stirred for an additional 5 min. Next, it was diluted with ether (30 mL) and washed with water (3 × 10 mL) and brine (10 mL), and the organic phase was dried over anhydrous Na_2_SO_4_. The removal of the solvent under vacuum conditions afforded **24** (94 mg, 88%) as a yellow syrup.

(4aR,9S,9aS)-9a-formyl-7-isopropyl-6-methoxy-1,1,4a-trimethyl-2,3,4,4a,9,9a-hexahydro-1H-fluoren-9-yl acetate (**25**). Ac_2_O (1 mL) and DMAP (5 mg) were added to a solution of **24** (41 mg, 0.13 mmol) in pyridine (2 mL) at 0 °C, and the reaction mixture was stirred at room temperature for 1 h, at which time TLC showed no starting material. Then, water (4 mL) was added to quench the reaction at 0 °C and the mixture was stirred for an additional 15 min. Next, ether (30 mL) was added, and the phases were shaken and separated. The organic phase was washed with 2 N HCl (5 × 10 mL), H_2_O (10 mL), sat aq NaHCO_3_ (5 × 10 mL), and brine (10 mL), and was dried over anhydrous Na_2_SO_4_. The removal of the solvent under vacuum conditions afforded a crude product, which was purified with a flash chromatography column on silica gel (20 % *t*-BuOMe/hexanes) to yield 40 mg of **25** (86 %) as a yellow syrup. [α]_D_^25^ = –46.1° (c = 1.0 CHCl_3_). ^1^H NMR (CDCl_3_, 500 MHz) δ: 0.87 (dd, *J* = 12.9, 5.8 Hz, 1 H), 0.94 (d, *J* = 7.1 Hz, 1 H), 1.17 (d, *J* = 6.9 Hz, 3 H), 1.18 (d, *J* = 6.9 Hz, 3 H), 1.20 (s, 3 H), 1.25 (s, 3 H), 1.38 (s, 3 H), 1.43–1.49 (m, 1 H), 1.52–1.73 (m, 2 H), 1.76–1.85 (m, 1 H), 2.10 (s, 3 H), 3.25 (h, *J* = 6.9 Hz, 1 H), 3.84 (s, 3 H), 6.53 (s, 1 H), 6.57 (s, 1 H), 7.06(s, 1 H), 9.62 (s, 1 H). ^13^C NMR (CDCl_3_, 125 MHz) δ: 18.2 (CH_2_), 21.4 (CH_3_) 22.6 (CH_3_), 22.7 (CH_3_), 26.0 (CH), 26.9 (2 x CH_3_), 27.9 (CH_3_), 29.7 (CH_2_), 35.2 (CH_2_), 38.5 (CH_2_), 47.4 (C), 55.5 (CH_3_), 77.2 (C), 78.5 (C), 102.5 (CH), 122.9 (CH), 129.8 (C), 136.4 (C), 150.3 (C), 158.1 (C), 170.9 (C), 204.5 (CH). IR (film): 1788, 1731 1482, 1442, 1327, 1258, 1221, 1049, 1078, 775 cm^−1^. HRMS (FAB) *m*/*z* calcd for C_23_H_32_O_4_Na (M + Na^+^) 395.2198, found 395.2210.

Treatment of aldol **24** with Amberlyst A-15: Obtention of compound **12**. Amberlyst A-15 (40 mg) was added to a solution of **24** (82 mg, 0.25 mmol) in CH_2_Cl_2_ (10 mL) and the mixture was stirred at room temperature for 6 h, at which time TLC showed no starting material. Then, it was filtered through a silica gel pad and washed with CH_2_Cl_2_ (50 mL). After the evaporation of the solvent under vacuum conditions, compound **12** (65 mg, 91 %) was obtained as a colorless syrup.

(4aR,9S,9aS)-7-isopropyl-6,9-dimethoxy-1,1,4a-trimethyl-2,3,4,4a,9,9a-hexahydro-1H-fluorene-9a-carbaldehyde (**26**). Sodium methoxide (22 mg, 0.41 mmol) was added to a solution of **11** (113 mg, 0.34 mmol) in MeOH (8 mL) at 0 °C, and the reaction mixture was stirred at this temperature for 15 min, at which time TLC showed no starting material. Then, HCl 2N (4 mL) was added at 0 °C to quench the reaction and the mixture was stirred for an additional 30 min. Next, it was diluted with ether (80 mL) and washed with water (3 × 15 mL) and brine (3 × 10 mL), and the organic phase was dried over anhydrous Na_2_SO_4_. The removal of the solvent under vacuum conditions afforded a crude product, which was purified with a flash chromatography column on silica gel (20 % *t*-BuOMe/hexanes) to yield **26** (101 mg, 86%) as a yellow syrup. [α]_D_^25^= –22.6° (c = 11.3 CHCl_3_). ^1^H NMR (CDCl_3_, 500 MHz) δ: 0.57 (s, 3 H), 1.14 (s, 3 H), 1.21 (d, *J* = 6.9 Hz, 3 H), 1.23 (d, *J* = 6.9 Hz, 3 H), 1.24–1.30 (m, 2 H), 1.32 (s, 3 H), 1.48–1.58 (m, 2 H), 1.81–1.87 (m, 2 H), 3.30 (h, *J* = 6.9 Hz, 1 H), 3.56 (s, 3 H), 3.85 (s, 3 H), 4.78 (s, 1 H), 6.58 (s, 1 H), 7.19 (s, 1 H), 9.85 (s, 1 H). ^13^C NMR (CDCl_3_, 125 MHz) δ: 18.3 (CH_2_), 22.78 (CH_3_), 22.84 (CH_3_), 22.85 (CH_3_), 22.86 (CH_3_), 27.7 (CH_3_), 28.8 (CH), 47.6 (C), 55.5 (CH_3_), 59.4 (CH_2_), 36.6 (CH_2_), 37.9 (CH_2_), 35.6 (C), 67.7 (C), 86.0 (CH), 102.7 (CH), 122.6 (CH), 132.1 (C), 135.3 (C), 150.6 (C), 157.7 (C), 206.9 (CH). IR (film): 1731, 1492, 1462, 1370, 1285, 1221, 1051, 1024, 772 cm^−1^. HRMS (FAB) *m*/*z* calcd for C_22_H_32_O_3_Na (M + Na^+^) 367.2249, found 367.2253.

Treatment of methoxyaldehyde **26** with Amberlyst A-15: Obtention of compound **12**.: Amberlyst A-15 (50 mg) was added to a solution of **26** (88 mg, 0.26 mmol) in CH_2_Cl_2_ (10 mL) and the mixture was stirred at room temperature for 8 h, at which time TLC showed no starting material. Then, it was filtered through a silica gel pad and washed with CH_2_Cl_2_ (50 mL). After the evaporation of the solvent under vacuum conditions, compound **12** (59 mg, 81 %) was obtained as a colorless syrup.

Treatment of dialdehyde **11** with 30% (+)-CSA: Obtention of compound **12**. CSA (70 mg, 0.3 mmol) was added to a solution of **11** (297 mg, 0.9 mmol) in CH_2_Cl_2_ (10 mL) and the mixture was stirred at room temperature for 60 h. Following the general procedure, 230 mg (90%) of compound **12** was obtained.

*O-*methyl taxodal (**27**). An O_3_/O_2_ mixture was slowly bubbled through a solution of **12** (243 mg, 0.86 mmol) in dry CH_2_Cl_2_ (15 mL) at −78 °C, and the course of the reaction was monitored by TLC. When the TLC showed no starting material, the solution was flushed with argon, triphenylphosphine (293 mg, 1.11 mmol) was added, and the mixture was stirred at room temperature for 4 h. Then, the solvent was removed, affording a crude product, which was purified with a flash chromatography column on silica gel (20% *t*-BuOMe /hexanes) to give pure **27** (244 mg, 90%) as a white solid. Mp=83 °C. [α]_D_^25^= –108.8° (c = 1.0 CHCl_3_). ^1^H NMR (CDCl_3_, 500 MHz) δ: 1.22 (d, *J* = 6.9 Hz, 3 H), 1.23 (d, *J* = 6.9 Hz, 3 H), 1.25 (s, 3 H), 1.29 (s, 3 H), 1.57–1.63 (m, 1 H), 1.67 (s, 3 H), 1.69–1.76 (m, 2 H), 1.99 (ddd, *J* = 13.6, 13.6, 3.6 Hz, 1 H), 2.20 (ddd, *J* = 13.2, 13.2, 3.8 Hz, 1 H), 2.44 (ddd, *J* = 13.4, 13.4, 3.8 Hz, 1 H), 3.29 (h, *J* = 6.9 Hz, 1 H), 3.93 (s, 3 H), 7.06 (s, 1 H), 7.61 (s, 1 H), 9.73 (s, 1 H). ^13^C NMR (CDCl_3_, 125 MHz) δ: 18.2 (CH_2_), 22.4 (CH_3_), 22.5 (CH_3_), 25.5 (CH_3_), 26.4 (CH), 26.4 (CH_3_), 27.1 (CH_3_), 38.1 (CH_2_), 40.3 (CH_2_), 44.6 (C), 53.2 (C), 55.6 (CH_3_), 109.9 (CH), 125.6 (C), 134.9 (C), 136.4 (CH), 147.9 (C), 161.3 (C), 191.5 (CH), 213.2 (C). IR (KBr): 1691, 1608, 1553, 1461, 1343, 1259, 1165, 997, 758 cm^−1^. HRMS (FAB) *m*/*z* calcd for C_20_H_28_O_3_Na (M + Na^+^) 339.1936, found 339.1950.

Taxodal (**28**). Potassium carbonate (69 mg, 0.5 mmol) and thiophenol (0.05 mL, 0.45 mmol) were added to a solution of **27** (96 mg, 0.30 mmol) in HMPA (10 mL), and the reaction mixture was heated at 160 °C for 10 min. Then, TLC showed no starting material, and the reaction mixture was cooled to room temperature and then extracted with ether (60 mL). The organic phase was washed with water (2 × 10 mL) and brine (2 × 10 mL) and was dried over anhydrous Na_2_SO_4_. The removal of the solvent under vacuum conditions afforded a crude product, which was directly purified with a flash chromatography column on silica gel (40% *t*-BuOMe/hexanes) to give taxodal (**28**) (75 mg, 83%) as a white foam. [α]_D_^25^= –10.3° (c = 0.5, CHCl_3_). ^1^H NMR (CDCl_3_, 500 MHz) δ: 1.18 (s, H), 1.23 (s, 3 H), 1.27 (d, *J* = 6.9 Hz, 3 H), 1.28 (d, *J* = 6.9 Hz, 3 H), 1.55–1.58 (s, 1 H), 1.56 (s, 3 H), 1.61–1.72 (m, 2 H), 1.97–2.02 (m, 1 H), 2.18 (ddd, *J* = 13.0, 13.0, 3.9 Hz, 1 H), 2.44 (ddd, *J* = 13.4, 13.4, 3.9 Hz, 1 H), 3.32 (h, *J* = 6.9 Hz, 1 H), 7.18 (s, 1 H), 7.74 (s, 1 H), 9.20 (br s, 1 H), 9.70 (s, 1 H).^13^C NMR (CDCl_3_, 125 MHz) δ: 19.5 (CH_2_), 22.70 (CH_3_), 22.74 (CH_3_), 25.9 (CH_3_), 27.5 (CH), 27.6 (CH_3_), 30.3 (CH_3_), 38.8 (CH_2_), 40.8 (CH_2_), 45.0 (C), 53.4 (C), 116.3 (CH), 126.7 (C), 133.3 (C), 138.3 (CH), 148.8 (C), 160.6 (CH_3_), 192.1 (CH), 215.1 (C). IR (film): 3260, 1683, 1612, 1579, 1275, 1172, 999, 759 cm^−1^. HRMS (FAB) *m*/*z* calcd for C_19_H_26_O_3_Na (M + Na^+^) 325.1870, found 325.1859.

*O-*methyl cupresol (**29**). 1-hexene (1.75 mL, 4 mmol), 10 mL of a solution of NaH_2_PO_4_ (5%), and sodium chlorite (102 mg, 1.14 mmol) were added to a solution of aldehyde **27** (141 mg, 0.45 mmol) in *t*-BuOH (15 mL), and the reaction mixture was stirred at room temperature for 40 h. At this time, TLC showed no starting material, and ether (60 mL) was added. The organic phase was washed with water (2 × 15) and brine, was dried over anhydrous Na_2_SO_4_, and was concentrated to give **29** (120 mg, 81 %) as a with foam. **29a** (α-hydroxy). ^1^H NMR (CDCl_3_, 500 MHz) δ: 1.164 (s, 3 H), 1.18 (d, *J* = 7.0 Hz, 3 H), 1.19 (d, *J* = 7.0 Hz, 3 H), 1.32 (s, 3 H), 1.40 (s, 3 H), 3.26 (h, *J* = 7.0 Hz, 1 H), 3.90 (s, 3 H), 6.65 (s, 1 H), 7.91 (s, 1 H). ^13^C NMR (CDCl_3_, 125 MHz) δ: 18.0 (CH_2_), 22.4 (CH_3_), 22.8 (CH_3_), 25.5 (CH), 25.70 (CH_3_), 25.75 (CH_3_), 26.71 (CH_3_), 31.5 (CH_2_), 37.9 (CH_2_), 39.3 (C), 43.6 (C), 55.54 (CH_3_), 104.1 (CH), 105.6 (C), 116.9 (C), 128.6 (CH), 135.9 (C), 149.4 (C), 161.99 (C), 164.56 (C). **29b** (β-hydroxy). ^1^H NMR (CDCl_3_, 500 MHz) δ: 0.67 (br s, 3 H), 1.159 (s, 3 H), 1.21 (d, *J* = 7.0 Hz, 3 H), 1.22 (d, *J* = 7.0 Hz, 3 H), 1.37 (br s, 3 H), 3.26 (h, *J* = 7.0 Hz, 1 H), 3.92 (s, 3 H), 6.77 (s, 1 H), 7.91 (s, 1 H). ^13^C NMR (CDCl_3_, 125 MHz) δ: 18.4 (CH_2_), 22.4 (2 x CH_3_), 25.5 (CH), 26.69 (CH_3_), 26.71 (CH_3_), 28.4 (CH_3_), 36.9 (CH_2_), 37.9 (CH_2_), 41.3 (C), 42.5 (C), 55.58 (CH_3_), 104.4 (CH), 105.6 (C), 116.9 (C), 128.6 (CH), 135.9 (C), 149.4 (C), 161.99 (C), 164.56 (C). HRMS (FAB) *m*/*z* calcd for C_20_H_28_O_4_Na (M + Na^+^) 355.1885, found 355.1897.

Cupresol (**30**). BBr_3_ (0.5 mL, 0.5 mmol) was added to a solution of **29** (60 mg, 0.18 mmol) in dry CH_2_Cl_2_ (12 mL) at 0 °C, and the reaction mixture was stirred at room temperature for 20 h, at which time TLC no showed starting material. Then, the reaction was cooled at 0 ° C, quenched with sat. aq. NaHCO_3_ (1 mL), and stirred for an additional 10 min. The mixture was diluted with ether (70 mL) and the organic phase was washed with water (3 × 15 mL) and brine (3 × 10 mL). The removal of the solvent under vacuum conditions afforded a crude product, which was purified with a flash chromatography column on silica gel (40% ether/hexanes) to yield **30** (50 mg, 87%) as a white foam. **30a** (α-hydroxy). ^1^H RMN (CDCl_3_, 500 MHz) δ: 0.70 (br s, 3 H), 1.163 (s, 3 H), 1.272 (d, *J* = 6.9 Hz, 3 H), 1.278 (d, *J* = 6.9 Hz, 3 H), 1.39 (s, 3 H), 3.04 (br s, 1 H), 3.16 (h, *J* = 7.0 Hz, 1 H), 5.37 (br s, 1 H), 6.61 (s, 1 H), 7.94 (s, 1 H). ^13^C RMN (CDCl_3_, 125 MHz) δ: 17.9 (CH_2_), 22.32 (CH_3_), 22.40 (CH_3_), 22.8 (CH_3_), 25.7 (CH_3_), 26.84 (CH_3_), 27.0 (CH), 31.3 (CH_2_), 36.8 (CH_2_), 39.3 (C), 41.2 (C), 109.9 (CH), 117.4 (C), 129.37 (C), 129.42 (CH), 133.3 (C), 146.4 (C), 158.3 (C), 164.5 (C). **30b** (β-hydroxy). ^1^H NMR (CDCl_3_, 500 MHz) δ: 0.70 (br s, 3 H), 1.158 (s, 3 H), 1.28 (d, *J* = 6.9 Hz, 3 H), 1.29 (d, *J* = 6.9 Hz, 3 H), 1.36 (br s, 3 H), 3.04 (br s, 1 H), 3.16 (h, *J* = 7.0 Hz, 1 H), 5.37 (br s, 1 H), 6.75 (s, 1 H), 7.95 (s, 1 H). ^13^C NMR (CDCl_3_, 125 MHz) δ: 18.3 (CH_2_), 22.34 (CH_3_), 22.35 (CH_3_), 22.87 (CH_3_), 24.8 (CH_3_), 25.4 (CH_3_), 26.7 (CH), 31.6 (CH_2_), 37.8 (CH_2_), 41.2 (C), 42.0 (C), 109.5 (CH), 117.7 (C), 129.37 (C), 129.37 (CH), 133.3 (C), 149.4 (C), 158.2 (C), 164.5 (C). HRMS (FAB) *m*/*z* calcd for C_19_H_26_O_4_Na (M + Na^+^) 341.1729, found 341.1735.

## 4. Conclusions

In summary, seco-abietane dialdehydes, which result from the oxidative rupture of the C6–C7 double bond of 6,7-dehydro-8,11,13-abietatriene diterpenes, are transformed under acidic conditions into the corresponding 4a-methyltetrahydrofluorene derivatives. This process could be proposed as a new biogenetic pathway from abietane diterpenes to taiwaniaquinoids. Using this unprecedented reaction, the first enantiospecific synthesis of taxodal (**28**) and cupresol (**30**) has been achieved.

## Data Availability

Data are contained within the article or Appendix A.

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
