# Peer review of "Unprecedented Elimination Reactions of Cyclic Aldols: A New Biosynthetic Pathway toward the Taiwaniaquinoid Skeleton"

_molecules, 2023, doi:10.3390/molecules28041524_

Round 1
Reviewer 1 Report
This manuscript reports a novel synthetic methodology to access taiwaniaquinoid derivatives by acid treatment of 6,7-seco-abietane dialdehydes. The reaction proceeds in high yields and selectivities. Furthermore, a range of 6,7-seco-abietane dialdehydes with different functionalities were evaluated and the reaction also occurred successfully. Authors also postulate a possible mechanism, which is in agreement with the experimental results. Finally, the syntheses of taxodal and cupresol were achieved.
The described reaction is quite interesting and it could have a high scope in the preparation of other diterpenoids. For this reason, I consider that this manuscript deserves to be published after minor revisions as followings.
1) If compounds 14, 16, 18 and 21 are known, their references should be included. If not, are their corresponding 6,7-dehydro-8,11,13-abietatriene diterpenes available or described in literature? If not, the preparation of these compounds should be described in the supporting information.
2) Page 7, Line 154, 30 mol % instead of 30%. The same in Scheme 5.
Author Response
- If compounds 14, 16, 18 and 21 are known, their references should be included. If not, are their corresponding 6,7-dehydro-8,11,13-abietatriene diterpenes available or described in literature? If not, the preparation of these compounds should be described in the supporting information.
We thank the reviewer for the comment. Seco-abietane dialdehydes 14, 16, 18 and 21 were obtained by reductive ozonolysis of the corresponding 6,7-dehydro-8,11,13-abietatriene derivatives. These compounds are a part of our current research and their preparation is described in J.J. Guardia, PhD Thesis, University of Granada, 2016. ISBN 9788491258643. We have included this reference on Table 2 for clarity.
- Page 7, Line 154, 30 mol % instead of 30%. The same in Scheme 5.
We thank the reviewer for the observation. We have corrected this typo in the revised manuscript.
Reviewer 2 Report
Guardia, Chahboun and co-workers reported an interesting elimination reaction of cyclic aldols, which may provide a new biosynthetic pathway toward the Taiwaniaquinoid skeleton in the future. This work was a continuation of their attempt to synthesize taiwaniaquinoids derivatives. The authors gave a very comprehensive background which included all relevant references in their Introduction section. Besides, the whole manuscript was well organized and the data were sufficient to support the authors’ conclusions. Thus, the manuscript is recommended to publish after revision. My comments are as follows:
1. Treatment of dialdehyde 5 under different acidic conditions resulted in phenol (6), aldehyde (7), and standishinal (8) with different yield ratio which is very interesting. Though the authors gave a possible mechanism for the transformation of dialdehyde 5 into phenol 6, the underlying mechanism of the acidity (or acid type)-dependent formation of the 6, 7, and 8 was not discussed.
2. Some corrections to the NMR data are needed.
3. Peak-picking, peak integration and assignment are recommended to provided in their 1H NMR spectra.
Author Response
1). Treatment of dialdehyde 5 under different acidic conditions resulted in phenol (6), aldehyde (7), and standishinal (8) with different yield ratio which is very interesting. Though the authors gave a possible mechanism for the transformation of dialdehyde 5 into phenol 6, the underlying mechanism of the acidity (or acid type)-dependent formation of the 6, 7, and 8 was not discussed.
We thank the reviewer for the comment. We have added a new synthetic scheme (Scheme 2) clarifying the mechanism for the transformation of dyaldehyde 5 into standishinal (8) and the subsequent dehydration to obtain compound 7.
2). Some corrections to the NMR data are needed.
We thank the reviewer for the observation. We have revised and amended some mistakes in the NMR data.
Reviewer 3 Report
In this manuscript, Juan and coworkers discovered a novel elimination reaction of cyclic aldols and applied this reaction to the enantio-specific synthesis of bioactive natural cupresol and taxodal. The study is interesting and would be appropriate to be published in Molecules after minor revisions.
1. Table 2 may be improved by presenting more results of the same substrates under different conditions.
2. Scheme 4, the structure of intermediate III is wrong the aldehyde part.
Author Response
1). Table 2 may be improved by presenting more results of the same substrates under different conditions.
We thank the reviewer for the comment. We have assay treatment of dialdehyde 11 under acidic conditions (resin Amberlyst A-15) and results were included in Table 2.
2). Scheme 4, the structure of intermediate III is wrong the aldehyde part
We thank the reviewer for the observation. We have corrected this typo in the revised manuscript.